# Mechanochemistry-driven engineering of 0D/3D heterostructure for designing highly luminescent Cs–Pb–Br perovskites

Kyeong-Yoon Baek [1,11], Woocheol Lee [1,9,11], Jonghoon Lee [1], Jaeyoung Kim [1], Heebeom Ahn[1], Jae Il Kim [2,3], Junwoo Kim [1,10], Hyungbin Lim[1], Jiwon Shin [1], Yoon-Joo Ko[4], Hyeon-Dong Lee[2], Richard H. Friend [5], Tae-Woo Lee [2,3,6✉], Jeongjae Lee [7✉], Keehoon Kang [2,3,8✉] & Takhee Lee [1,8✉]

Embedding metal-halide perovskite particles within an insulating host matrix has proven to be an effective strategy for revealing the outstanding luminescence properties of perovskites as an emerging class of light emitters. Particularly, unexpected bright green emission observed in a nominally pure zero-dimensional cesium–lead–bromide perovskite ($Cs_4PbBr_6$) has triggered intensive research in better understanding the serendipitous incorporation of emissive guest species within the $Cs_4PbBr_6$ host. However, a limited controllability over such heterostructural configurations in conventional solution-based synthesis methods has limited the degree of freedom in designing synthesis routes for accessing different structural and compositional configurations of these host–guest species. In this study, we provide means of enhancing the luminescence properties in the nominal $Cs_4PbBr_6$ powder through a guided heterostructural configuration engineering enabled by solid-state mechanochemical synthesis. Realized by an in-depth study on time-dependent evaluation of optical and structural properties during the synthesis of $Cs_4PbBr_6$, our target-designed synthesis protocol to promote the endotaxial formation of $Cs_4PbBr_6/CsPbBr_3$ heterostructures provides key insights for understanding and designing kinetics-guided syntheses of highly luminescent perovskite emitters for light-emitting applications.

[1] Department of Physics and Astronomy, Seoul National University, Seoul 08826, Korea. [2] Department of Materials Science and Engineering, Seoul National University, Seoul 08826, Korea. [3] Research Institute of Advanced Materials, Seoul National University, Seoul 08826, Korea. [4] Laboratory of Nuclear Magnetic Resonance, National Center for Inter-University Research Facilities, Seoul National University, Seoul 08826, Korea. [5] Cavendish Laboratory, University of Cambridge, Cambridge CB3 0HE, UK. [6] School of Chemical and Biological Engineering, Institute of Engineering Research, Soft Foundry, Seoul National University, Seoul 08826, Korea. [7] School of Earth and Environmental Sciences, Seoul National University, Seoul 08826, Korea. [8] Institute of Applied Physics, Seoul National University, Seoul 08826, Korea. [9] Present address: R&D Center, SK Hynix Inc., Icheon-si, Gyeonggi-do 17336, Korea. [10] Present address: Samsung Electronics Co., Pyeongtaek-si, Gyeonggi-do 17786, Korea. [11] These authors contributed equally: Kyeong-Yoon Baek, Woocheol Lee. ✉email: twlees@snu.ac.kr; jl635@snu.ac.kr; keehoon.kang@snu.ac.kr; tlee@snu.ac.kr

In the time span of only a few years, lead halide perovskites with the general formula of APbX₃ (A and X being a monovalent cation and halide anion, respectively) have established superior performance in many key optical and electronic devices, including solar cells[1–3], light-emitting diodes[4–8], photodetectors[9], and memory devices[10]. Particularly, Cs-based perovskites have emerged as a new class of emitters because of their high quantum yield, narrow full width at half maximum (FWHM), and versatility to widely tune the bandgap in the visible-light range[11,12]. Notwithstanding their promising aspects as next-generation optoelectronic material, lead halide perovskites critically suffer from degradation under atmospheric environments. Hence, in order to extend the stability of these perovskite emitters, various methods such as encapsulating them with organic ligands[13] or embedding them in low dimensional inorganic perovskite scaffold of similar chemical compositions has been introduced[14–18]; while use of organic ligands can confer additional benefits in terms of potential surface functionalization, use of solid inorganic matrix is often more desirable in terms of atmospheric stability.

Among low dimensional networked perovskites in the Cs–Pb–X compositional space, $Cs_4PbX_6$ (referred to as the zero-dimensional (0D) phase) adopts a structure in which the $[PbX_6]^{4-}$ octahedra are detached from each other by surrounding $Cs^+$ atoms, whereas conventional three-dimensional (3D) $CsPbX_3$ perovskites consist of corner-sharing octahedra with $Cs^+$ cations filling the cavities[13,19,20]. While the 0D phase shows superior stability compared to the 3D perovskite phase, visible-light emission from the 0D phase itself was not expected to occur due to the large intrinsic bandgap of 0D structure which corresponds to the ultraviolet (UV) region[18–22]. Thus, experimentally observed green emission from 0D $Cs_4PbBr_6$ has remained a puzzle and conflicting theories on the exact origin of such emission exist: while some have attributed the improved stability and high photoluminescence (PL) quantum efficiency (QE) to the serendipitous formation of a 0D/3D host–guest structure with 3D perovskite embedded within the insulating 0D matrix[14,23–29], others have predicted the existence of intrinsic defects within the 0D $Cs_4PbBr_6$ phase as the origin of such PL[21,22,30–32]. Although a definitive agreement has not been reached, recent experimental investigations using Raman spectroscopy[25], neutron diffraction[26], cathodoluminescence imaging[27], and superfluorescence emission[28] have provided evidence in favor of unintended inclusion of 3D $CsPbBr_3$ nano-sized emitters in the 0D $Cs_4PbBr_6$ matrix. These investigations have subsequently kindled a series of studies on the formation mechanism of such embedded 3D emitters by real-time tracking of PL signals during solution-based synthesis of 0D phases[33,34]. However, the inability to fully "freeze" the reaction during solution-based synthesis methods has hampered a detailed analysis of the underlying mechanism responsible for the formation of 3D/0D heterostructures. Furthermore, a controllable growth of such heterointerfaces is challenging in solution phase due to its homogeneous nature, which in turn limits the controllability over the number of possible final heterostructures, as well as the synthetic routes towards them.

In contrast to solution-based chemistry, mechanochemical synthesis (MCS) has recently received increasing attention as an alternative and facile method to produce halide perovskites[35–41]. MCS, or ball milling, is a synthetic process in which solid-state reactions are induced by the mechanical energy of rotating balls in the reaction container without the need for (often toxic) solvent media. Crucially, and to our advantage, MCS reactions can be halted at any point during synthesis by simply turning off the mechanical agitation[42–44]. This allows us to capture the intermediate reaction stages involved in the formation of the final product. Moreover, the MCS process has the additional advantage of being able to use different solid phases[37], as long as they are of

the same stoichiometry (e.g., $4CsBr + PbBr_2$ and $CsPbBr_3 + 3$ $CsBr$ are all expected to yield $Cs_4PbBr_6$), whereas such flexibility is absent in solution-based synthesis methods.

Exploiting this synthetic degree of freedom of MCS, we provide means of enhancing the green emission in the 0D $Cs_4PbBr_6$ phase through an in-depth study of time-dependent structural and optical properties of various co-existing Cs–Pb–Br polymorphs formed during a slow MCS process. Surprisingly, intermediate-stage samples obtained during the MCS process exhibited significantly higher PL intensity than the final 0D $Cs_4PbBr_6$ product, the bright emission of which is shown to arise from an endotaxial formation of 3D $CsPbBr_3$ particles inside the 0D $Cs_4PbBr_6$ matrix. Furthermore, we exploit mechanochemistry as a route to accessing diverse heterostructural 0D/3D host–guest configurations (i.e., both in composition and structure) and providing insights into the kinetics underlying the formation mechanism of such embedded heterostructures. This understanding allows us to target-design a synthesis protocol to facilitate the formation of 0D phase components on the surface of the parent 3D phase in an endotaxial manner, thereby resulting in a two-fold enhancement in the PLQE values and ultimately providing guidelines for the large-scale synthesis of highly luminescent perovskite emitters. We also demonstrate the possibility to further improve the emissivity of these heterostructures by ligand encapsulation, which may prove crucial for facile processing in terms of device production.

## Results

**Mechanochemically synthesized Cs–Pb–Br composite perovskites.** In order to devise any synthesis in an informed manner, the underlying mechanism and its driving forces need to be understood first. Since these considerations in turn depend on the exact technique used in practice, we first explore the available methods for realizing synthesis through mechanochemical means. For MCS, several methods can be employed such as hand-grinding (through mortar and pestle), rotating mill-type, planetary-type, and shaker-type ball mill[35,42,45]. To conduct a systematic observation of time-dependent properties of synthesized perovskite compounds, the following conditions should be satisfied: (i) a uniform force should be applied to the powder throughout the entire synthesis process and (ii) the temperature of the mixture must be maintained at a constant value to implement a quasistatic process. The rotating mill-type MCS method can satisfy these conditions and, more importantly, the slower kinetics of this process allows us to observe the intermediate phases present within the conversion from the precursors to the 0D $Cs_4PbBr_6$ phase (the full reaction takes 48 hr to complete; further details are discussed in Supplementary Notes 1.1 and 1.2). Figure 1 illustrates the rotating mill-type MCS used to synthesize Cs–Pb–Br composite perovskites. Precursor salts of CsBr and $PbBr_2$, shown in the bottom–left part of Fig. 1, were put into a cylindrical alumina jar with a 10-cm diameter and 15-cm height (see photo in the bottom–right part of Fig. 1) along with stainless-steel grinding balls. The MCS process is conducted by rotating the jar with two mechanical rollers. Through this process, three different phases of 0D $Cs_4PbBr_6$, 3D $CsPbBr_3$, and two-dimensional (2D) $CsPb_2Br_5$ can be synthesized by adjusting the stoichiometric ratio of the precursors (CsBr:PbBr₂ = 4:1, 1:1, and 1:2 for 0D, 3D, and 2D material, respectively). The phase purity of mechanochemically synthesized powder samples was confirmed via powder X-ray diffraction (PXRD) patterns (Supplementary Fig. 2). As depicted in the upper-right part of Fig. 1, 3D $CsPbBr_3$ and 0D $Cs_4PbBr_6$ are composed of corner-sharing and isolated $[PbBr_6]^{4-}$ octahedra, respectively, whereas 2D $CsPb_2Br_5$ is composed of $[Pb_2Br_5]^-$ polyhedron layers with alternating $Cs^+$ atoms. The entire process was conducted in ambient conditions at room

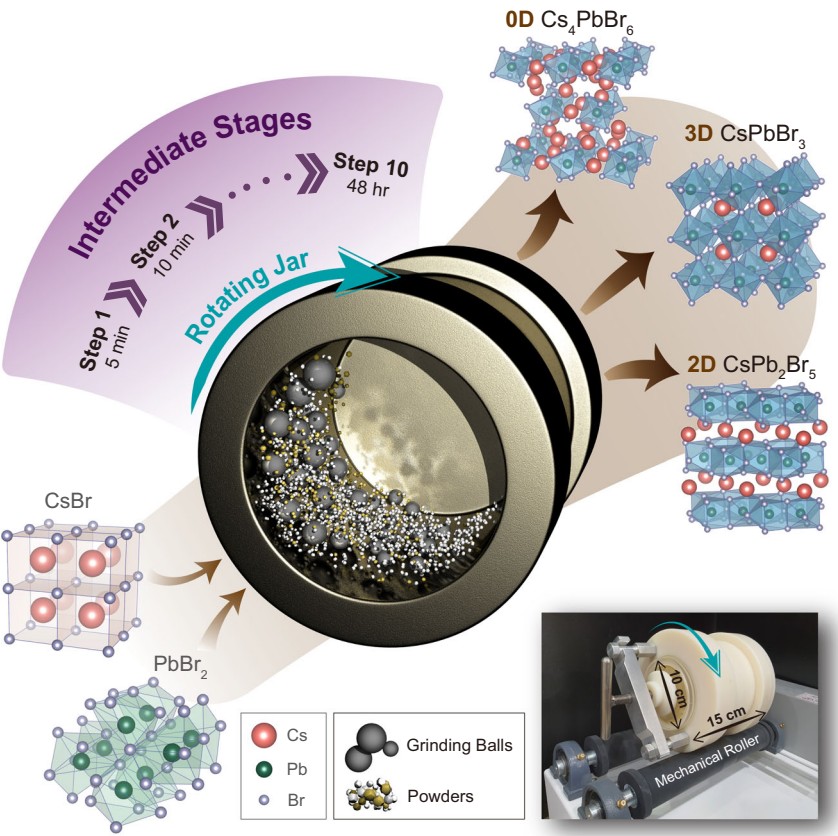

**Fig. 1 Rotating mill-type MCS process.** Schematic illustration of the overall MCS procedure for the synthesis of perovskite phases. Two precursor salts (CsBr and PbBr$_2$) and grinding balls are put into a container (alumina jar) and rotated continuously. This process results in different Cs–Pb–Br perovskite phases of 0D Cs$_4$PbBr$_6$, 3D CsPbBr$_3$, and 2D CsPb$_2$Br$_5$. The slow reaction nature of rotating mill-type MCS enables the subdivision of intermediate process into several steps, which allows precise observation of intermediate phases. Crystal structures of precursors and each perovskite phase are schematically presented. The figure in the bottom–right shows a photograph of our experimental setup.

temperature, which demonstrates the simple and practical nature of the process when compared with the relatively strict conditions required for conventional synthesis routes of metal–halide perovskites[37,46,47]. The pure-phase Cs–Pb–Br composite perovskites, as well as their stable intermediate phases, can be systematically synthesized and analyzed by capitalizing on the controllability of the rotating mill-type MCS process. Specifically, the MCS process enables us to study in full by "mapping" the intermediate products at different reaction time coordinates (i.e., milling times).

**Time evolution of optical properties during the MCS process.** We began the MCS process with CsBr and PbBr$_2$ precursors mixed in a stoichiometric ratio of 4:1 to form 0D Cs$_4$PbBr$_6$ perovskite powder. The top image in Fig. 2a shows the synthesized perovskite powder samples under ordinary fluorescent light during the MCS process as a function of milling time, from the start to 2880 min (48 hr). The hue of the products gradually changed from orange to white. Considering that the colors of the two precursors, CsBr and PbBr$_2$, are white and 3D CsPbBr$_3$ is the only powder with orange color among all possible Cs–Pb–Br perovskite phases (0D, 2D, and 3D), only 5 min of milling time is deemed sufficient to initiate the formation of 3D CsPbBr$_3$ (ref. [39,42]). Powder samples with white colors (observed after ~360 min of milling time) indicate that the 0D Cs$_4$PbBr$_6$ perovskite is the dominant phase in these samples. Further verification and quantification of the intermediate phases will be discussed in detail later.

Along with the visual changes in hue, the evolution of the PL intensity under a UV lamp (wavelength of 365 nm) showed that the PL intensity first increased, maxing out at a milling time of 180 min, and then decreased steadily until the completion of reaction (bottom image of Fig. 2a and solid black symbols in Fig. 2b). When PLQE values of samples in each milling step were measured through densely packing the powder into a quartz cell with a width of 2 mm (see Supplementary Note 1.3 for detailed notes on the PLQE acquisition of powder samples), a similar trend of reaching the maximum value for the sample in the intermediate stage was observed (open cyan symbols in Fig. 2b). The slight discrepancy in the maximum conditions of the milling time between the PL intensity and PLQE (180 min and 360 min, respectively) is attributed to a relatively low absorption for the 360-min sample in comparison with that for the 180-min sample (see the onset elevation difference in the dotted black box of Fig. 2d). We note that sample before milling showed a near zero PLQE value of 0.07 %, indicating that no emissive properties were present beforehand (see Supplementary Fig. 5). The FWHM of the PL peak reached its minimum (18 nm) at a milling time of 360 min and then gradually increased afterward (Fig. 2c and Supplementary Fig. 6). From these results, an optimum milling time ranging from 180 to 360 min could be identified in terms of both PL intensity and PL peak FWHM of powder samples in the intermediate stages of the synthesis.

Absorption spectrum for each milling step is shown in Fig. 2d. In the early stages of the MCS process (until a milling time of ~90 min), band-to-band absorption with an onset of ~532 nm

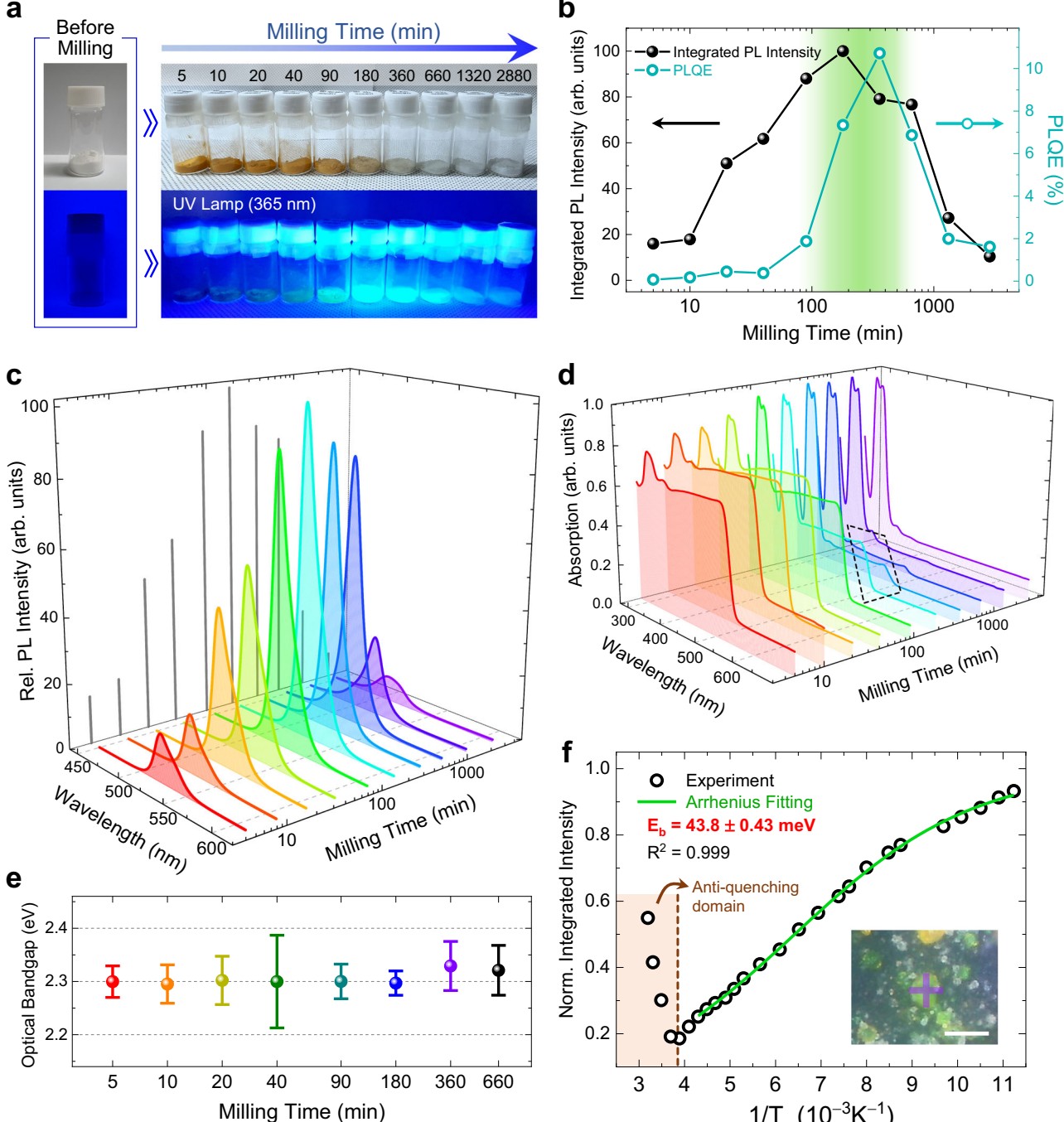

**Fig. 2 Time evolution of optical properties and the origin of highly luminescent green emission during the MCS process. a** Images of powder samples during the MCS process with successive increase in milling time. The samples show a gradual change in hue under ordinary fluorescent (top) and in intensity under UV (bottom) lamps. **b** Time evolution of integrated PL intensity (solid black symbols) and PLQE (open cyan symbols) with an excitation wavelength of 365 nm. The green gradient marks the maximum region of both emission parameters. **c** Time evolution of PL spectra as milling time elapses with an excitation wavelength of 365 nm. The gray lines in the milling time–rel. PL intensity plane indicate the orthogonal projection of the PL spectra. **d** Time evolution of absorption spectra as milling time elapses. The dashed black box indicates the onset elevation difference between 180-min and 360-min samples near a wavelength of 532 nm. **e** Calculated optical bandgaps as a function of milling time. Values of optical bandgaps correspond to the absorption onsets of the Tauc plots[60] as shown in Supplementary Fig. 8. Error bars are derived from the deviations in linear fitting. **f** Temperature-dependent integrated PL intensity of the green emissive region in the 90-min sample (one of the intermediate-stage samples with high intensity green emission) under a 405-nm laser excitation source. The inset shows an optical image of the target scene with laser collimation marked in semi-transparent purple cross. Scale bar, 20 μm. Source data are provided as a Source Data file.

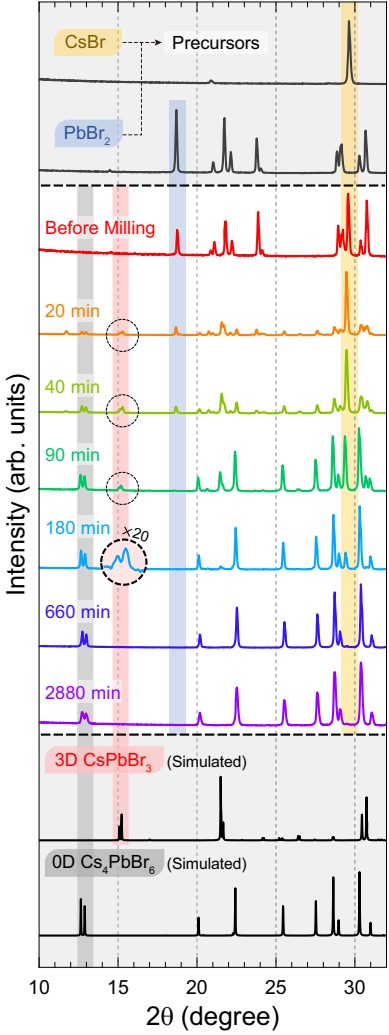

**Fig. 3 Time evolution of perovskite structures through PXRD.** PXRD patterns of CsBr and PbBr₂ (top gray panel), samples from different milling times (middle panel), and reference patterns for 3D CsPbBr₃ (COD ID 4510745; ref. [47]) and 0D Cs₄PbBr₆ (COD ID 4002857; ref. [22]) (bottom gray panel). Representative peaks of CsBr, PbBr₂, CsPbBr₃, and Cs₄PbBr₆ are marked by semi-transparent yellow, blue, red, and black strips, respectively. The full diffraction patterns are given in Supplementary Fig. 18. Source data are provided as a Source Data file.

was dominant and an optical bandgap of ~2.30 eV could be derived through the Tauc plot (Supplementary Fig. 8) with its ratio between absorption and scattering constants ($\alpha$) obtained by the Kubelka–Munk function[48]. The extracted optical bandgap was nearly constant throughout the synthesis process, as shown in Fig. 2e. Note that the onset position of ~532 nm coincides with the characteristic absorption for 3D CsPbBr₃ (ref. [49].), which further supports the presence of 3D phases within the samples. As the milling process continued, a noticeable absorption peak around ~320 nm (corresponding to ~3.87 eV) emerged, which indicates the formation of 0D Cs₄PbBr₆ (ref. [19]), whereas the aforementioned band absorption at ~532 nm gradually diminished (Fig. 2d and Supplementary Fig. 9). These results imply that the 3D phase is abundant in the early stages of synthesis and the 0D phase becomes dominant as the milling time increases toward completion. Hence, we can expect that the 0D Cs₄PbBr₆ phase is formed at the expense of the 3D CsPbBr₃ phase at the later stages of the MCS process; this initial mechanistic insight is discussed in further detail below.

In order to ascribe the origin of such high intensity green emission on intermediate stages, the exciton binding energy, a quantity that depends on the nature of the perovskite phase (i.e., structure and size), was obtained by variable-temperature PL measurements. Assuming that the decrease in PL intensity upon heating is solely due to the increased thermal dissociation rate of excitons, the exciton binding energy could be obtained from an Arrhenius-type relation, $I(T) \propto 1/(1 + A \exp(-E_b/(k_B T)))$, where $I(T)$ is the integrated PL intensity at temperature $T$, $E_b$ is the exciton binding energy, and $k_B$ is the Boltzmann constant[50]. A small exciton binding energy of $43.8 \pm 0.4$ meV for the 90-min sample, which is in agreement with previously reported values for 3D CsPbBr₃ (ref. [51]), was determined from Fig. 2f and implies Wannier–Mott excitonic property. Among the various forms of CsPbBr₃ (e.g., powder, single crystal, and nanocrystal), the powder and single crystal forms of CsPbBr₃ exhibit very poor PLQE (less than 0.1 %), with almost no detection of PL[21,49]. This suggests that the green luminescent moiety observed in our samples (Supplementary Fig. 11) contains nanocrystalline CsPbBr₃. Moreover, the presence of a distinct Arrhenius-type region at low temperatures and the "anti-quenching" region at high temperatures (above ~250 K) suggest the presence of heterostructure where 3D CsPbBr₃ nanocrystals (NCs) are embedded in 0D Cs₄PbBr₆ matrix, as previously reported[29]. On the other hand, weak intensity green emission in the final stage of the synthesis exhibited an exciton binding energy of $225.7 \pm 18.6$ meV (Supplementary Fig. 14) which lies on the tightly bound Frenkel excitonic property[21,22]. Hence, these results imply that the high intensity emission in the intermediate synthesis steps can be attributed to the presence of luminescent 3D CsPbBr₃ NCs embedded within the 0D Cs₄PbBr₆ host. Further details on microscopic spatial identification of emissive moiety are described in Supplementary Note 2.

**Time evolution of Cs–Pb–Br perovskite phases and reaction kinetics during the MCS process.** The evolution of optical properties discussed above must correlate with the evolution of crystalline structures during the MCS process, which we now discuss. Figure 3 shows PXRD data of synthesized perovskite powder samples as a function of MCS milling time. The time evolution of the PXRD patterns (Fig. 3 and Supplementary Fig. 18) shows the clear appearance of 3D CsPbBr₃ as an intermediate product as well as the targeted formation of 0D Cs₄PbBr₆ during synthesis. At the early steps of synthesis, peaks corresponding to the two precursors (CsBr and PbBr₂) are observed (each marked with semi-transparent yellow and blue strips, respectively, in Fig. 3). As the milling time increases, the peaks corresponding to the 3D CsPbBr₃ initially grow in intensity and then subsequently diminish (see the semi-transparent red strip from the 20 to 180-min panel in Fig. 3). Concurrently, the peaks corresponding to 0D Cs₄PbBr₆ steadily increase (see the semi-transparent black strip from the 20-min panel in Fig. 3). Note that the small intensity peak at a low angle diffraction of 11.7° for the 20-min and 40-min sample arises from the 2D CsPb₂Br₅ phase, which can be further supported by the following NMR analysis. We note that this small amount of 2D CsPb₂Br₅ does not influence the green PL emission (see Supplementary Note 3.3).

To reveal the presence of any ternary or amorphous compound other than 3D CsPbBr₃ and 0D Cs₄PbBr₆ that can be corroborated with the PXRD results, magic-angle-spinning (MAS) solid-state nuclear magnetic resonance (ssNMR), a sensitive characterization tool used to determine the local environments to an atomic level, was performed. The ¹³³Cs MAS ssNMR signals exhibit superior spectral resolution and sensitivity when compared with their ²⁰⁷Pb and ⁷⁹Br counterparts[31,39,41,52]. Thus, we focused on characterizing

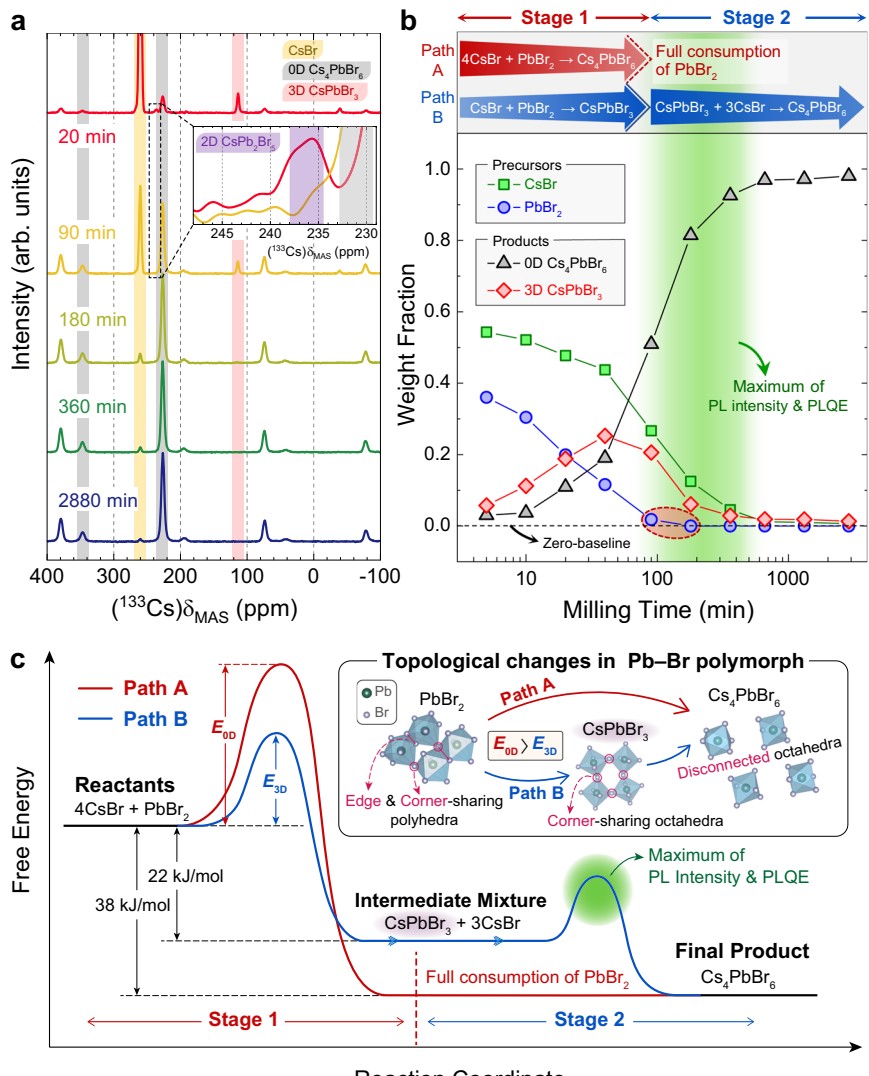

**Fig. 4 Compositional evolution of Cs–Pb–Br polymorphs and the underlying kinetics of competing reaction paths. a** $^{133}$Cs MAS ssNMR spectra of samples with different milling times at 10 kHz MAS and 300 K. CsBr, $Cs_4PbBr_6$, $CsPbBr_3$, and $CsPb_2Br_5$ peaks are marked with semi-transparent yellow, black, red, and purple strips, respectively. The remaining peaks correspond to spinning side bands. Inset shows magnified spectra of 20 and 90-min samples (red and yellow line, respectively). The full spectra of **a** are given in Supplementary Fig. 19. **b** Weight fractions of precursors and synthesized Cs–Pb–Br perovskites for each sample calculated through Rietveld refinement process. The same graph shown in molar fraction can be found in Supplementary Fig. 21. The dividing of stages coincides with the near consumption of $PbBr_2$ (below 2 wt%) and different reaction paths contributing to these stages are labeled as Path A and Path B. The green gradient is marked at the same region as in Fig. 2b. **c** A schematic reaction kinetics diagram of two competing paths (Paths A and B) in the synthesis of 0D $Cs_4PbBr_6$ from the precursors. Formation enthalpy of $CsPbBr_3$ (22 kJ/mol) and $Cs_4PbBr_6$ (38 kJ/mol) are from ref. [24]. The inset illustrates the potential origin of difference between the two activation energies of $E_{0D}$ and $E_{3D}$. Source data are provided as a Source Data file.

the $^{133}$Cs signals. Figure 4a shows a series of peaks for different phases that contain Cs polymorphs. The four peaks marked with semi-transparent yellow, black, and red strips in this figure correspond to precursor CsBr ($\delta_{iso} = 260.2$ ppm), 0D $Cs_4PbBr_6$ ($\delta_{iso} = 226.5$ ppm, 346.7 ppm), and 3D $CsPbBr_3$ ($\delta_{iso} = 113.5$ ppm), respectively[31,39]. In accordance with PXRD data, 3D $CsPbBr_3$ appeared as an intermediate product and noticeably diminished in the later steps of synthesis. Moreover, the presence of 2D $CsPb_2Br_5$ was detected only in the 20-min sample (i.e., in the early stage of the synthesis) with its $\delta_{iso}$ at 235.7 ppm (see the semi-transparent purple strip in the inset of Fig. 4a) and vanishes afterwards. The spectrum of the 2880-min sample showed no signs of 3D $CsPbBr_3$ peaks, indicating that this final product contains a negligible level of 3D phase, below the detection limit of NMR (see Supplementary Fig. 19).

As we confirmed that the major constituents participating in the synthesis are the two precursors (CsBr and $PbBr_2$) and the 0D and 3D perovskite phases, the weight fraction for each constituent in the samples at each step is quantified from PXRD results of Fig. 3 using Rietveld refinement analysis (Fig. 4b). Based on the dominant reactants that contribute to the formation of 0D $Cs_4PbBr_6$, the synthesis process can be tentatively divided into two stages (Stage 1: 0−90 min; Stage 2: 180−2880 min). In Stage 1, the two precursors react to form 0D $Cs_4PbBr_6$ in parallel with 3D $CsPbBr_3$ whereas in Stage 2, after the full consumption of $PbBr_2$ (see the zero-baseline and red dashed oval in Fig. 4b), the as-synthesized 3D $CsPbBr_3$ formed in Stage 1 begins to take part in the production of 0D $Cs_4PbBr_6$ with the remaining CsBr (reaction formulae occurring in each stage are indicated by red and blue arrows in the upper part of Fig. 4b). Note that the

maxima of the PL intensity and PLQE occurred at milling times of ~180 to ~360 min and these milling times correspond to the early period of Stage 2 (see the green gradients in Figs. 2b and 4b).

From these observations, the overall reaction kinetics regarding this synthesis process can be understood as schematically illustrated in Fig. 4c. Starting from CsBr and $PbBr_2$ with a stoichiometric ratio of 4:1, the existence of 3D $CsPbBr_3$-abundant stages (until ~90 min of milling shown in Fig. 4b) implies the possibility of two different routes in forming 0D $Cs_4PbBr_6$. In one path (named Path A and marked by a red line in Fig. 4c) the two precursors react directly to form 0D $Cs_4PbBr_6$ with the activation energy of $E_{0D}$, whereas in the other path (named Path B and marked by a blue line in Fig. 4c) the precursors initially form a 3D $CsPbBr_3$ intermediate with the corresponding activation energy of $E_{3D}$, that subsequently reacts with the remaining CsBr to produce 0D $Cs_4PbBr_6$. Here, both Path A and Path B occur simultaneously during the synthesis process, as long as the reactants (CsBr and $PbBr_2$) are present in the mixture. Thus, the reaction produces both 3D $CsPbBr_3$ and 0D $Cs_4PbBr_6$ at early stages of the synthesis. However, the fact that these two reaction pathways occur at the same time and share the same reactants means that they are in competition against each other; hence, the relative kinetics between the two pathways is the key factor in determining which pathway dominates the reaction. Out of the two competing reaction paths, Path B is expected to be dominant as observed from the time-dependent fractions of each phase (Fig. 4b), because the 3D phase is formed before the 0D phase (Stage 1) and the rapid growth of 0D phase happens later in Stage 2. This implies that the activation energy $E_{3D}$, required to form 3D $CsPbBr_3$, is lower than $E_{0D}$ as depicted in Fig. 4c. The discrepancy between $E_{0D}$ and $E_{3D}$ can be explained by the difference in the degree to which the topological connectivity of Pb–Br polymorphs changes among different crystalline phases during the formation process (see the inset of Fig. 4c). We speculate that Path B requires less energy and is thus more favorable than Path A since the initial stage of reaction Path B involves a structural rearrangement of Pb–Br polyhedra, from the edge and corner-sharing polyhedra ($PbBr_2$) to simply corner-sharing octahedra ($CsPbBr_3$), whereas Path A accompanies a complete break-up of the interlinked Pb–Br polyhedra into isolated $[PbBr_6]^{4-}$ octahedra ($Cs_4PbBr_6$). Correlating this reaction kinetics analysis with the observed emissive stages provides a valuable picture for understanding and maximizing the PL properties of the mechanochemically synthesized Cs–Pb–Br perovskites.

**Designing a synthesis route for highly emissive Cs–Pb–Br perovskite**. The correlation of the reaction kinetics governing conversion from 3D $CsPbBr_3$ to 0D $Cs_4PbBr_6$ with the identification of the "3D-embedded 0D" configuration indicates that the inward formation of 0D from the surface of pre-formed 3D (i.e., endotaxial growth of 0D from 3D) is a key synthesis strategy for enhancing PL. To our advantage, MCS can directly employ the solid 3D phase as the starting ingredient for synthesizing Cs–Pb–Br perovskite composites with the optimum composition. Moreover, because the MCS reaction is fundamentally a heterogeneous process involving the physical contact of two particles, reactions are typically expected to initiate from particle surfaces, thus bestowing the opportunity to guide synthesis from any given particle in an endotaxial manner. Thus, a synthesis recipe for maximizing PL intensity can be designed by selecting 3D $CsPbBr_3$ and CsBr as the starting ingredients (denoted as Case 1 in Fig. 5a) for facilitating endotaxial growth of 0D $Cs_4PbBr_6$ from the parent 3D $CsPbBr_3$. By contrast, starting compositions not amenable to such endotaxial growth are expected to result in powder products

with different structural configurations albeit the same chemical composition as Case 1. Thus, two other synthesis routes with CsBr and $PbBr_2$ (denoted as Case 2; i.e., a similar reaction process as outlined from Fig. 1 to Fig. 4 but with a different compositional ratio of starting ingredients) and 0D $Cs_4PbBr_6$ and $PbBr_2$ (denoted as Case 3, wherein endotaxial growth of 3D is expected from the parent 0D) as the starting reagents were employed as controls. As the PL and PLQE of synthesized samples depend on the compositional ratio of 3D $CsPbBr_3$ and 0D $Cs_4PbBr_6$ (see the correlation between Figs. 2b and 4b), we first found the optimum nominal wt% of 3D $CsPbBr_3$ in the final product for achieving the highest PL intensity and PLQE; this value was ~7 wt% (see Supplementary Note 4.2). This intentional under-stoichiometry of CsBr (i.e., $x < 3$ in Case 1; Fig. 5a) is expected to result in the favorable formation of a 0D/3D host–guest system in which unreacted 3D particles are embedded in the 0D matrix rather than a pure-phase 0D product. To confirm our prediction, final products with the optimum composition of ~7 wt% 3D $CsPbBr_3$ were synthesized via the aforementioned synthesis routes (Cases 1, 2, and 3) according to the stoichiometry outlined in Fig. 5a. While the PXRD patterns along with the compositional analysis by Rietveld refinement show ~7 wt% of 3D $CsPbBr_3$ phase in the final products for all three cases as expected (see Supplementary Fig. 29), the corresponding PL and PLQE data show a marked difference in their emissive properties (box-and-whisker plot of Fig. 5b and Supplementary Fig. 30): for both PL intensity and PLQE, Case 1 was the most superior, Case 3 was the least, and Case 2 was in between. This result can be understood by different spatial distribution within the $CsPbBr_3$–$Cs_4PbBr_6$ heterostructure: despite being the same chemical composition, the final product in Case 1 is likely to have more $CsPbBr_3$ NCs embedded within the $Cs_4PbBr_6$ matrix, which provides a stable emissive environment for $CsPbBr_3$ particles (see the illustration in the upper panel of Fig. 5c). By contrast, the final product in Case 3 is likely to have a majority of $CsPbBr_3$ particles residing on the exterior of $Cs_4PbBr_6$ because the reaction is designed to occur from the surface of the parent $Cs_4PbBr_6$ (see the illustration in the lower panel of Fig. 5c). $CsPbBr_3$ particles formed on the exterior becomes either highly susceptible to degradation or agglomerate into bulk particles, which in both cases rendering them as weakly emissive moities[49]. In line with this scenario, $^{133}Cs$ MAS ssNMR analyses of these emissive samples show additional shoulder-like features with slightly reduced shifts to the main 0D Cs peaks (see Supplementary Note 4.4). We tentatively assign this feature to the 0D Cs atoms residing in the interface of 0D $Cs_4PbBr_6$ and 3D $CsPbBr_3$, therefore supporting the encapsulation hypothesis as described above.

The structural analysis of these contrasting final products (Cases 1 and 3) using selected area electron diffraction (SAED) and high-resolution transmission electron microscopy (HRTEM) further confirms the predictions illustrated in Fig. 5c. The SAED pattern in Case 1 clearly shows a mixture of diffraction peaks originating from 3D and 0D phases, whereas only the 0D phase is visible in the SAED pattern of Case 3 (Fig. 5d, e; projected 1D-profile in Fig. 5f). Consistent with this result, HRTEM image of Case-1 powder shows a mixture of hexagonal and orthorhombic lattice fringes corresponding to the 0D and 3D phases, respectively (Fig. 5g). Examples of such phases are shown by the cyan ($0D_I$) and magenta ($3D_I$) boxes together with fast Fourier transformed (FFT) images obtained from the magnified HRTEM images (right images of Fig. 5g). FFT images of other regions can be found in Supplementary Fig. 34. By contrast, no such features related to the 3D $CsPbBr_3$ phase could be found in the corresponding HRTEM images for the Case 3 sample (Supplementary Note 4.5), indicating the absence of 3D/0D heterostructures, which is consistent with the poor emission observed in this sample.

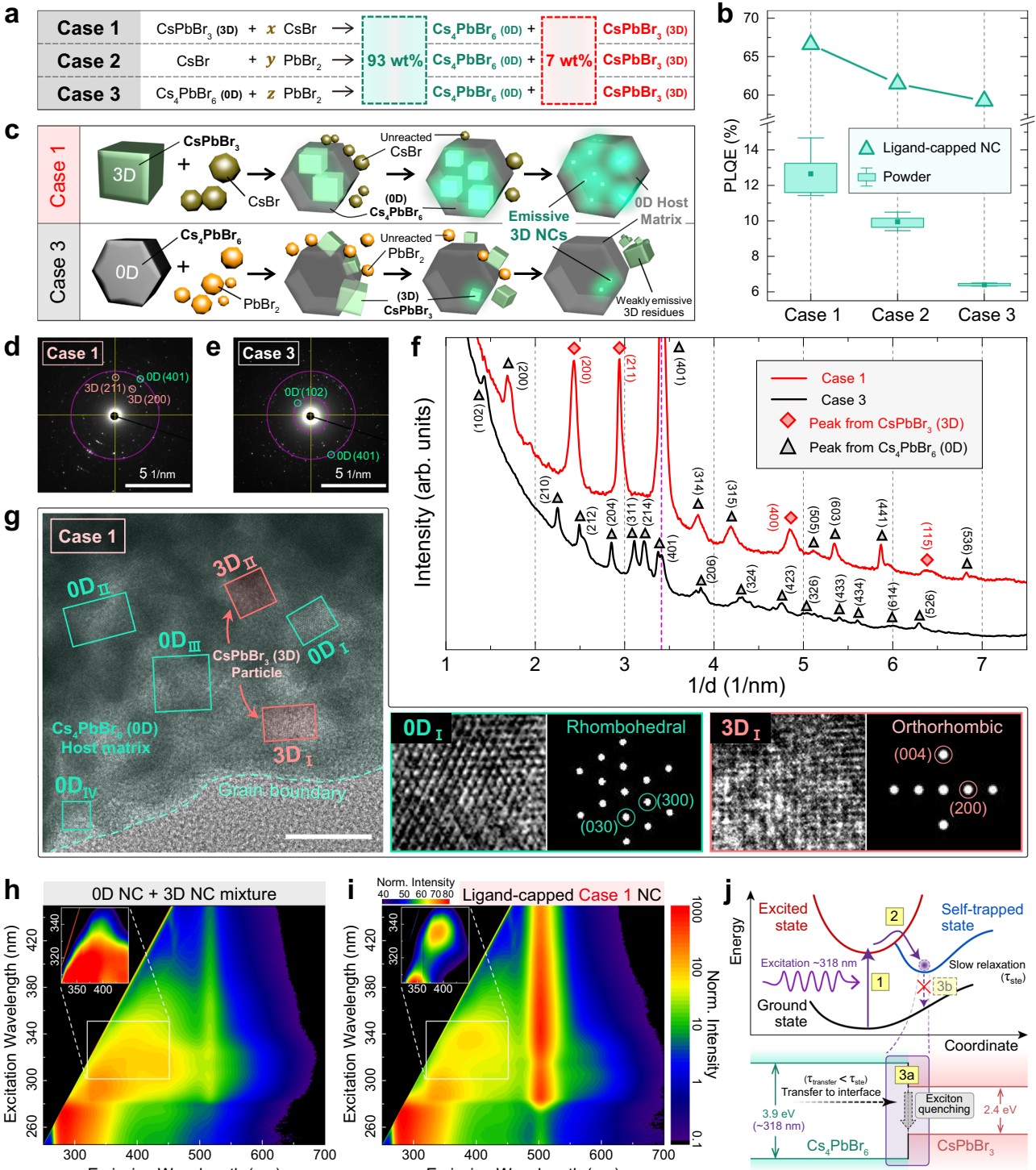

**Fig. 5 Target-designed synthesis route for enhancing emissivity of the CsPbBr₃–Cs₄PbBr₆ heterostructure. a** Three different reaction schemes (Cases 1, 2, and 3) for producing the same final CsPbBr₃–Cs₄PbBr₆ stoichiometries but with different initial precursor compositions. **b** PLQE for Cases 1, 2, and 3 as in powder form (box-and-whisker symbols) and ligand-capped NC form (triangle symbols) under excitation wavelength of 365 nm and 405 nm, respectively. Samples in powder form were dispersed in toluene for measurements. **c** Schematic illustration regarding the evolution of heterostructural samples for Cases 1 and 3. Case 1 shows the formation of 0D Cs₄PbBr₆ from the surface of 3D CsPbBr₃, whereas Case 3 shows the formation of 3D CsPbBr₃ from the surface of 0D Cs₄PbBr₆. **d–f** SAED patterns of the final products from Case 1 (**d**) and Case 3 (**e**), and graph of the rotational average intensity at a distance (= 1/d) from the center of each pattern (**f**). **g** HRTEM image of the final product from Case-1 powder. The left image of **g** is covered with semi-transparent false colors of cyan to indicate the 0D Cs₄PbBr₆ matrix and magenta to indicate the 3D CsPbBr₃ particles. FFT images (right images in each box of **g**) were obtained from the magnified HRTEM images (left images in each box of **g**). Scale bar, 20 nm. **h, i** Contour plots of PL intensity as a function of excitation and emission wavelengths from the simple mixture of 0D NC and 3D NC (**h**) and ligand-capped Case-1 NC (**i**). The inset of **h** and **i** are rescaled colormaps near the excitation wavelength of 318 nm. **j** Schematic illustration explaining PL quenching in the inset of **i** at an excitation wavelength with an energy of the bandgap of 0D Cs₄PbBr₆ (3.9 eV; ~318 nm). Slow relaxation of self-trapped excitons leads to exciton quenching in the interface of 0D and 3D. Source data are provided as a Source Data file.

**Ligand encapsulation of MCS product through wet-milling.** For practical application of these perovskites to optoelectronic devices, deposition in forms of perovskite films is generally required. While this can be achieved by several means, large-scale production of uniform films is often performed by printing the perovskite ink onto a substrate. For this, the ball-milled product needs to be further encapsulated with organic ligands to ensure the colloidal stability of NC suspension. To this end, each powder sample of Cases 1, 2, and 3 was engineered into NCs via wet-milling process with oleylamine (see Methods for milling procedures) and the PLQE of each case increased relatively to that of each powder form with Case-1 sample exhibiting the highest value of 66.6% (triangle symbols of Fig. 5b, see Supplementary Note 4.6). In order to ensure that the host–guest configuration of the sample is maintained after the wet-milling process, we observed the PL intensity as a function of excitation and emission wavelengths. Figure 5h shows the contour map of the PL intensity from a simple mixture of ligand-capped 0D and 3D NCs, (i.e., a physical mixture with no direct interface between the two phases). Emission near 510 nm at the excitation range of 280 nm and above is attributed to 3D NC and the rest is attributed to 0D NC (see Supplementary Note 4.7). From the Case-1 NC sample as shown in Fig. 5i, most area of the plot can be attributed to either 3D or 0D. However, a marked difference exists between the simple mixture and the Case-1 NC sample near the excitation wavelength of 318 nm (see the insets of Fig. 5h and i): quenching of PL emission is observed near the 318-nm excitation for Case-1 NC sample, whereas the mixture sample does not exhibit such effect. We ascribe this difference to the existence of 3D and 0D interface for Case-1 NC, as schematically illustrated in Fig. 5j. When excited with a wavelength of 318 nm (~3.9 eV) which matches with the bandgap of 0D $Cs_4PbBr_6$, self-trapped excitons are readily formed which possess a slower relaxation time for radiative recombination than the free excitons[53] (i.e., following the sequential process marked as '1' and '2' yellow boxes in Fig. 5j). Meanwhile, the as-formed excitons travel towards the interface of 0D and 3D, and are expected to be quenched due to an ineffective energy transfer at the 0D/3D interface at room temperature[18,20,54] (i.e., following the '3a' process rather than the '3b' process in Fig. 5j). Hence, the observed quenching of 318-nm emission in the ligand-encapsulated Case-1 NC indicates the persistence of 3D/0D interfaces, which implies that the host–guest configuration of Case-1 sample is maintained throughout the wet-milling process.

## Discussion

In this work, we demonstrated an efficient synthesis route for producing highly luminescent host–guest configuration driven by tracing the time evolution of different Cs–Pb–Br phases and the emissive properties during the formation of 0D $Cs_4PbBr_6$. By exploiting the MCS process, we observe that the formation of the 3D $CsPbBr_3$ phase precedes the growth of the 0D $Cs_4PbBr_6$ phase and that the delayed formation of $Cs_4PbBr_6$ in an endotaxial manner on the pre-formed $CsPbBr_3$ particle surfaces coincides with the green emission observed in the samples. While the precise origin of this emission has remained controversial, our current results using a variety of structural and optical characterization techniques favor the "$CsPbBr_3$-embedded $Cs_4PbBr_6$" structural motif as the key to achieving such bright green emission in the intermediate reaction stages. More importantly, understanding the mechanism of particle growth and emission origin, combined with the capacity of predictive solid-state synthesis in heterostructural systems bestowed by MCS, enabled us to clearly identify the strategy for producing highly luminescent perovskite emitters with host–guest heterostructural configuration.

The analytical framework built in this study can be easily expanded to a wide range of perovskite-related materials due to the simplicity of MCS procedures and the superior advantage of bypassing any solubility issues. Moreover, the highly emissive $CsPbBr_3$–$Cs_4PbBr_6$ products obtained from optimizing the MCS process may possess the immediate potential for use as new precursors to create ligand-capped nanocrystalline heterostructures through a reduction in size by further ball milling. Such a "top–down" approach to NC synthesis is a promising synthesis method due to its facile nature and high throughput[12,40,45,55]. Ligand capping of $CsPbBr_3$-embedded $Cs_4PbBr_6$ nanoparticles could result in inorganic–organic hybrid protective layers wherein the inorganic $Cs_4PbBr_6$ provides a stable shell for encapsulating emissive $CsPbBr_3$ particles and organic ligands provide a further barrier to degradation. Furthermore, while a simple oleyl ligand was used for NC encapsulation in the current work, we note that our approach of producing NC suspensions for film applications can be straightforwardly extended to accommodate various other functional ligands as widely exploited in recent perovskite light-emitting diodes and photovoltaic studies[13]. As discussed above, this functionalization of MCS-prepared NCs can open up possibilities for e.g., better atmospheric protection, enhanced charge conductivity, and improved colloidal formulation for large-scale printing of perovskite optoelectronics. Considering the potential of the MCS process as a synthesis method for mass-producing a wide range of host–guest (e.g., core–shell) type NCs with minimal use of toxic solvents, our study provides a novel perspective on large-scale synthesis and functionalization of metal–halide perovskites for light-emitting applications.

## Methods

### Sample preparation
*Mechanochemical synthesis of Cs–Pb–Br perovskites.* For all reactants involved in the synthesis of this paper, precursor salts of cesium bromide (CsBr, 99.9%, Sigma-Aldrich) and lead(II) bromide ($PbBr_2$, 98%, Sigma-Aldrich) were used. As-received CsBr salt was hand-ground using an agate mortar and pestle before each synthesis to match its particle size with that of the $PbBr_2$ salt. For the powder samples used for the measurements in Figs. 2–4, 100 mmol (21.281 g) of CsBr and 25 mmol (9.175 g) of $PbBr_2$ were put into an alumina jar along with a total of 336 g of stainless-steel grinding balls (4 balls 1.9 cm in diameter and 200 balls 0.6 cm in diameter). The alumina jar with the mixture was rotated in ball mill equipment (BD4530, LK Lab) with a rotating speed of 350 rpm with respect to the mechanical roller. In Cases 1, 2, and 3 of Fig. 5a, the precursors were added in the desired stoichiometric ratio to obtain 7 g of the final product along with a total of 400 g of stainless-steel balls (2 balls 1.9 cm in diameter, 2 balls 1.5 cm in diameter, 10 balls 1.2 cm in diameter, 20 balls 0.9 cm in diameter, and 150 balls 0.6 cm in diameter) and the duration time of the synthesis was equally set to 48 hours. For all the syntheses, the mass ratio of stainless-steel balls to precursor salts was set to be over 10 (refs. [35,44]). All syntheses were conducted in an ambient atmosphere and at room temperature.

*Ligand-capped NCs via wet-milling.* Wet-milling procedures for producing ligand-capped Case-1, Case-2, and Case-3 NCs shown in Fig. 5 follows protocols from literature[45] with minor modification. To specify, bulk powder of each case with 0.1 g in mass was initially loaded into a glass vial with 5 g of zirconia grinding balls (0.2 cm in diameter), giving fine-powdered samples. As-prepared powder sample was loaded into a glass vial with zirconia balls (50 μm in diameter), 0.5 g of oleylamine, and 10 mL of toluene (99.5%, Sigma-Aldrich). The vial was mounted on a lab rolling mixer (custom-made) and the rotating speed was set to 250 rpm with respect to the mechanical roller and the mixing time was set to 48 hours. After the mixing, a green suspension was obtained, which was diluted with 2 mL of toluene and precipitated with centrifugation at 11,515 × *g* for 5 minutes. The precipitate was redispersed in 2 mL of toluene and centrifuged at 1036 × *g* for 5 minutes to remove large sized particles. The final precipitate was redispersed in 2 mL of toluene.

### Optical characterization
*Photoluminescence.* Steady-state PL emission and PL excitation spectra were measured with a spectrofluorometer (JASCO FP-8500) utilizing a Xenon arc lamp with a power of 150 W. PLQE was measured using the same spectrofluorometer equipped with a 100 mm integrating sphere (ILF-835) coated with barium sulfate and the value was calculated using Jasco SpectraManager II software. A correction

for taking account of absorption from the scattered light (i.e., indirect configuration) was performed in order to rule out the possibility of overestimating the PLQE when measuring highly scattering powder samples[56,57]. Confocal PL mapping was performed using an XperRam 200 (Nanobase Inc.) instrument with a 405-nm laser excitation source and a diffraction-limited laser spot size of ~1 μm. Temperature-dependent PL measurements were conducted using the same equipment (XperRam 200, Nanobase Inc.), with the temperature controlled by liquid $N_2$.

*UV–Visible absorption.* Absorption spectra were obtained through reflectance measurements using a UV/Vis spectrophotometer (JASCO V-770) equipped with a 150-mm integrating sphere (ILN-925) coated with barium sulfate.

### Structural characterization

*Powder X-ray diffraction.* PXRD measurements were performed with Rigaku SmartLab diffractometer using a Cu Kα radiation source with a wavelength at 1.5406 Å. Phase fractions from the PXRD patterns were calculated by Rietveld refinement using GSAS-II software[58]. The simulated PXRD results (Fig. 3) were obtained using VESTA software[59] with CIF files from the crystallography open database[22,47].

*Solid-state NMR.* Solid-state $^{133}Cs$ ssNMR spectra were obtained using 4 mm cross-polarization (CP)-MAS probes and a 500 MHz Bruker ADVANCE III HD NMR spectrometer with 10 kHz spinning speed. A simple pulse-acquire sequence was used with flip angle of 20 degrees (52 kHz radiofrequency amplitude). Approximately 100 mg samples in mass were packed in disposable Kel-F inserts to mitigate issues with spinning instability.

### Electron microscopy observation

*Field emission SEM.* Synthesized perovskite powder samples were placed on a carbon tape. SEM images were obtained using a JSM-7800F (JEOL Ltd.) Prime equipped with SDD-type EDS operating at an acceleration voltage of 5 kV. EDS mapping images were obtained using the same equipment at an acceleration voltage of 10 kV.

*TEM and SAED.* TEM images were obtained with a JEM-F200 (JEOL Ltd.) operating at an acceleration voltage of 200 kV. Synthesized perovskite powder samples were dispersed in toluene and then dropped onto copper mesh coated with a carbon layer and Formvar film. SAED patterns were obtained using an aperture of 10 μm and an area of 150 nm.

### Data availability

The authors declare that the data supporting the findings of this study are available within the paper and its Supplementary Information file. Source data are provided with this paper.

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

## Acknowledgements
The authors appreciate the financial support of the National Research Foundation of Korea (NRF) grant (No. 2021R1A2C3004783, No. 2021R1C1C1010266, and No. 2016R1A3B1908431), the BrainLink program (No. 2022H1D3A3A01077343), and the Nano•Material Technology Development Program (No. 2021M3H4A1A02049651) through NRF funded by the Ministry of Science and ICT (MSIT) of Korea, and the industry–university cooperation program by Samsung Electronics Co., Ltd. (IO201211-08047-01). Jeongjae L. was supported by the NRF grant funded by MSIT of Korea (No. 2019R1A6A1A10073437). K.-Y.B. and H.L. acknowledge support from the Student-Directed Education (Undergraduate Research) Program at the Faculty of Liberal Education, Seoul National University (2021). The authors acknowledge support from the National Center for Inter-University Research Facilities (NCIRF) at Seoul National University for PXRD and NMR measurements and electron microscope observations, and Research Institute of Advanced Materials (RIAM) at Seoul National University for absorption measurements.

## Author contributions
K.-Y.B., W.L., Jeongjae L., K.K. and T.L. conceived and designed the experiments. Mechanochemically synthesized sample preparation was performed by K.-Y.B. and ligand-capped NC preparation was performed by J.I.K. Optical measurements were conducted and analyzed by K.-Y.B., Jonghoon L. and Jaeyoung K. with support from H.L., J.S., J.I.K., and H.-D.L. PXRD measurements and electron microscopy observations were conducted by K.-Y.B. with support from H.A., Junwoo K. and H.L. Weight fractions were calculated by W.L. NMR experiments were designed and analyzed by Jeongjae L. and Y.-J.K. K.K., Jeongjae L. and R.H.F. advised the overall mechanistic discussions on reaction kinetics and endotaxial growth. T.-W.L. led the discussion on the optical characterizations. T.L. supervised this research. K.-Y.B. designed and arranged the overall figures. The manuscript was prepared by K.-Y.B., W.L., T.-W.L., Jeongjae L., K.K. and T.L. with input from all authors.

## Competing interests
The authors declare no competing interests.
