## [Peer Review File · Nature Communications]

REVIEWER COMMENTS

Reviewer #1 (Remarks to the Author):

The manuscript by Baek et al. reported a mechanochemical method to synthesize highly emissive Cs–Pb–Br composite perovskite. By carrying out an in-depth study on time-dependent evaluation of optical and structural properties during the milling procedures, they demonstrated that the “CsPbBr₃-embedded Cs₄PbBr₆” structural motif are critical to achieve bright green emission in the intermediate reaction stages. Moreover, the kinetics underlying the formation of such embedded heterostructures was clearly disclosed based on series of time evolution tracking of XRD, PL, absorption, HRTEM, etc. Overall, the results shown in this work are very comprehensive and analytical framework may be expanded to a wide range of perovskite related materials. Therefore, I can recommend publication of this manuscript in Nature Communications after the following issues have been fully addressed.

1. In Fig. 2b, the as-synthesized Cs-Pb-Br complex shows an optimal PLQE of ~10%. This value is low compared to the generally reported CsPbBr₃/Cs₄PbBr₆ composites (PLQE, 40%-90%). The authors should clarify more on this relatively low PLQE.

2. For the reaction kinetics diagram proposed in this paper, it seems like the intermediate product CsPbBr₃ are expected to be solely observed during the mechanochemical synthesis. However, the time dependent XRD in Fig. 3a didn't verify this. The XRD peaks for both CsPbBr₃ and Cs₄PbBr₆ emerged at the early stage of ball milling (stage 1, 20 min), suggesting that 0D and 3D perovskites formed at the same time. In other words, there might be no such intermediate state. The authors may need to comment on this.

3. Following my last comment, the XRD patterns at 20 min and 40 min show a low angle diffraction near 11.5°, which can't be assigned to any of the materials mentioned in this work. This diffraction peak disappeared after 90 min. The authors should provide more explanations.

Reviewer #2 (Remarks to the Author):

Dear Authors,

I enjoyed reading your manuscript on the mechano-chemical synthesis of Cs-Pb-Br materials - in particular your detailed analysis of the evolution of materials formation. Indeed I find your work novel and refreshing as it provides some additional insight into the origin of green luminescence observed from what I agree are probably CsPbBr₃ inclusions in a higher bandgap matrix. Your correlative XRD and PL results show that the evolution of PL and XRD signatures indeed correlate and that there are also effects of changes in the PLQY to consider showing that PL is not the best quantitative probe of the amount of CsPbBr₃ formed. The manuscript is very well written and can in principle be published as-is. I would like the authors to consider the following suggestions for minor amendments or things to consider:

1) Consider to also add the PL and absorption spectra of samples before milling in Figure 2. This should of course be a fairly boring flat line but would confirm, that before milling no emissive species is present.

2) As noted by M Green and co-workers in a very comprehensive publication in 2015 (*J. Phys. Chem. Lett.* 2015, 6, 4774–4785), in materials with modest exciton binding energies, the excitonic absorption at the absorption onset obscures the actual bandgap. The bandgap cannot be directly determined from Tauc plots and authors should maybe consider to just refer to this as the absorption onset.

Responses to the Reviewer #1's comments

[Overall evaluation]

The manuscript by Baek et al. reported a mechanochemical method to synthesize highly emissive Cs–Pb–Br composite perovskite. By carrying out an in-depth study on time-dependent evaluation of optical and structural properties during the milling procedures, they demonstrated that the “CsPbBr₃-embedded Cs₄PbBr₆” structural motif are critical to achieve bright green emission in the intermediate reaction stages. Moreover, the kinetics underlying the formation of such embedded heterostructures was clearly disclosed based on series of time evolution tracking of XRD, PL, absorption, HRTEM, etc. Overall, the results shown in this work are very comprehensive and analytical framework may be expanded to a wide range of perovskite related materials. Therefore, I can recommend publication of this manuscript in Nature Communications after the following issues have been fully addressed.

[Response]

We appreciate the reviewer's valuable time and efforts for evaluating our manuscript. In the following, we summarized the point-by-point responses and changes we have made on the basis of the reviewer's comments and suggestions. Please note that we added additional remarks on page 6 of this response letter file suggesting a modification on the manuscript title and an additional minor correction that we found during this round of revision.

[Comment 1]

1. In Fig. 2b, the as-synthesized Cs–Pb–Br complex shows an optimal PLQE of ~10%. This value is low compared to the generally reported CsPbBr₃/Cs₄PbBr₆ composites (PLQE, 40%–90%). The authors should clarify more on this relatively low PLQE.

[Response]

We thank the reviewer's comments. As the reviewer commented, our highest internal PLQE value of 10.7 % from the 360-min powder sample (Fig. 2b) is low compared to generally reported values of 40–90 %. However, we note that our ligand-capped CsPbBr₃/Cs₄PbBr₆ species exhibits a PLQE value of 66.6 % (Fig. 4b) which is within the range of reported PLQE values as mentioned by the reviewer.

In the following, we would like to suggest the potential reasons behind the low observed PLQE values of the *powder* samples relative to the literature values. In short, our powder samples are (1) prepared under a synthesis approach different with those to the previous reports in terms of sample

washing and synthesis methods; (2) in addition, deviations in protocols for measuring PLQE values of powder samples may also contribute to the discrepancies.

1. Sample preparation and post-treatment methods:

- a. DMSO washing: Many reports present PLQE values for the *solid* samples after washing the as-formed product with dimethyl sulfoxide (DMSO) or isopropanol. For instance, PLQE value of 45 % was achieved only after washing the sample with DMSO [R1, R2] whereas only ~7% PLQE was achieved before the washing process. Hence, our PLQE values for the powder samples are comparable with reference values for the samples in their as-synthesized form (i.e., before any washing process). Here, enhancement of PLQE values via DMSO washing can be justified by dissolution of eliminating the weakly emissive 3D CsPbBr₃ bulk particles. Furthermore, depending on the amount of DMSO used for washing, PLQE values can vary within a large range of 60–95% [R3]. However, the PLQE data in Fig. 2b were measured in as-synthesized forms since our main focus was to track the time evolution of intrinsic properties of powder samples induced only through mechanochemical reactions (i.e., no intermixing effects due to the DMSO washing).
- b. Incomplete synthesis by MCS: As presented in Fig. 3c, unreacted precursors (CsBr and PbBr₂) coexist with the synthesized Cs–Pb–Br polymorphs before the completion of synthesis (e.g., 4.5 wt% of CsBr coexist with Cs₄PbBr₆/CsPbBr₃ mixture in 360-min sample) which might jeopardize the PLQE values. In addition, since MCS involves chemical reactions proceeding from the surface of the powder by mechanical friction, it may be prone to formation of defect sites that induce PL quenching and thereby result in lowering the PLQE values.

2. Accurate PLQE measurement protocol: Although this may constitute a minor effect towards the PLQE values, we outline here the importance of the measurement protocol in accurately characterizing the PLQE of the powder samples. When comparing PLQE values between different research groups, one needs to check whether the correction for sample scattering is taken into account or not (i.e., the ‘direct’ and ‘indirect’ configurations with regards to sample excitation and emission; see Methods section of the manuscript for details). Specifically, when measuring solid-state samples which scatter a fair proportion of the incident light, PLQE determination becomes susceptible to overestimation if the corresponding correction is not considered. In this regard, our PLQE measurement of powder samples were carefully calibrated following the protocol from ref. [R4] where emission induced from indirect excitation is subtracted from the as-measured emission, thereby eliminating the possibility of overestimating PLQE values especially for powder samples.

To sum up our response for the reviewer's comments, careful measures taken to minimize the extrinsic effects in experimental determination of PLQE values, along with different powder preparation strategies, likely contribute to the low (~10 %) PLQE values of the *powder* samples relative to the literature reports. Also, we would like to emphasize that the *absolute* values of PLQE in powder samples is not the main point of concern in this part of our study, where the *relative* evolution of PLQE values are more relevant for systematically investigating the synthesis mechanism. In contrast, measurements on the ligand-capped nanocrystal solutions (Fig. 4b), where the measurement is free from the above extrinsic factors, give the highest PLQE value of 66.6 %, which is well within the range of the value mentioned by the reviewer.

In response to the reviewer's comments, we revised the manuscript as following.

1. We added a new section (**Section 1.3. PLQE measurement of powder samples**) in the revised Supplementary Information which covers the explanation mentioned above.
2. We revised the following paragraph: "...When PLQE values of samples in each milling step were measured through densely packing the powder into a quartz cell with a width of 2 mm (see **Section 1.3 of the Supplementary Information for detailed notes on the PLQE acquisition of powder samples**), a similar trend of reaching the maximum value for the sample in the intermediate stage was observed..." on page 8, line 142 in the revised manuscript.

[Comment 2]

2. For the reaction kinetics diagram proposed in this paper, it seems like the intermediate product $CsPbBr_3$ are expected to be solely observed during the mechanochemical synthesis. However, the time dependent XRD in Fig. 3a didn't verify this. The XRD peaks for both $CsPbBr_3$ and Cs_4PbBr_6 emerged at the early stage of ball milling (stage 1, 20 min), suggesting that 0D and 3D perovskites formed at the same time. In other words, there might be no such intermediate state. The authors may need to comment on this.

[Response]

We appreciate the reviewer for the comment. Regarding this comment, we would like to stress that both Path A and Path B (Fig. 3d) occur *simultaneously* during the *whole* synthesis process, as long as the reactants ($CsBr$ and $PbBr_2$) are present in the mixture. Thus, the reaction produces both 3D $CsPbBr_3$ and 0D Cs_4PbBr_6 at early stages of the synthesis (where both of the reactants are present in abundance).

However, the fact that these two reaction pathways occurring at the same time (and sharing the same reactants) means that they are in competition against each other; hence, the relative *kinetics* between the two pathways is the key factor in determining which pathway dominates the reaction. Since the activation energy to form 0D Cs_4PbBr_6 (E_{0D}) is likely to be higher than that of the 3D $CsPbBr_3$ intermediate (E_{3D}),

3D CsPbBr₃ is formed at a faster rate than 0D Cs₄PbBr₆ (see their relative weight fractions during Stage 1 in Fig. 3c). This makes Path B (which involves 3D CsPbBr₃ as the intermediate product) the kinetically more favored pathway than Path A (which proceeds directly to form 0D Cs₄PbBr₆ as the final product). Therefore, we propose that most of PbBr₂ is initially consumed to form 3D CsPbBr₃ in the early stage (Stage 1), after which the 3D CsPbBr₃ (i.e., the only Br source when PbBr₂ precursor is absent) and the remaining CsBr precursor react to undergo a complete conversion to 0D Cs₄PbBr₆ (Stage 2).

In response to the reviewer's comment, we revised the manuscript as following.

1. We added new paragraph, “Here, both Path A and Path B occur simultaneously during the synthesis process, as long as the reactants (CsBr and PbBr₂) are present in the mixture. Thus, the reaction produces both 3D CsPbBr₃ and 0D Cs₄PbBr₆ at early stages of the synthesis. However, the fact that these two reaction pathways occur at the same time and share the same reactants means that they are in competition against each other; hence, the relative kinetics between the two pathways is the key factor in determining which pathway dominates the reaction.” on page 13, line 243 of the revised manuscript.
2. We note that the term ‘Intermediate Product’ in Fig. 3d can be misleading. So, we changed the terminology to ‘Intermediate Mixture’ in Fig. 3d.

[Comment 3]

3. Following my last comment, the XRD patterns at 20 min and 40 min show a low angle diffraction near 11.5°, which can't be assigned to any of the materials mentioned in this work. This diffraction peak disappeared after 90 min. The authors should provide more explanations.

[Response]

We thank the reviewer for the careful scrutiny of our manuscript. We have indeed overlooked the diffraction peak near 11.5°. **Figure R1** below is a zoomed-in plot near 11.5° from Fig. 3a. As shown in **Fig. R1**, we have confirmed that this peak is located at a precise angle of 11.74°, which originates from the (002) plane of 2D CsPb₂Br₅. In accordance with the reviewer's comment, this peak is observed only for 20-min and 40-min sample. This peak is not present before the milling (i.e., 0 min) and it vanishes at 90-min sample. This result is consistent with our NMR result provided in Fig. 3b where the inset shows evidence of 2D CsPb₂Br₅ presence at 20-min sample but not at 90-min sample.

In the original manuscript, it was erroneously mentioned that the presence of 2D CsPb₂Br₅ was only observable through NMR and was not confirmed by PXRD. We modified the text of the manuscript regarding this point which is detailed below. Here, it is worth mentioning (as also did from the manuscript) that Section 3.3 of the Supplementary Information file demonstrates the insignificant role of intermediate-

produced 2D CsPb₂Br₅ to the green PL emission of our samples. In addition, the calculated reference peaks of 2D CsPb₂Br₅ was presented in Supplementary Fig. 2c.

Figure R1. Powder XRD patterns of 0-min, 20-min, 40-min, and 90-min samples and reference pattern for 2D CsPb₂Br₅ (ref. [R5]). Diffraction peak at 11.74° marked as dashed black line corresponds to the (002) plane of 2D CsPb₂Br₅. Experimental data (top white panel) are magnified by 3 times and calculated data (bottom gray panel) is reduced by 5 times.

In response to the reviewer’s comment, we revised our manuscript as following.

1. We added a new sentence: “Note that the small intensity peak at a low angle diffraction of 11.7° for the 20-min and 40-min sample arises from the 2D CsPb₂Br₅ phase, which can be further supported by the following NMR analysis. We note that this small amount of 2D CsPb₂Br₅ does not influence the green PL emission (see Section 3.3 of the Supplementary Information).” on page 11, line 201 in the revised manuscript.

2. We changed the following original sentence “To reveal the presence of any ternary or amorphous compound other than 3D CsPbBr₃ and 0D Cs₄PbBr₆ that cannot be detected by PXRD, magic-angle-spinning (MAS) solid-state nuclear magnetic resonance (ssNMR), a sensitive characterization tool used to determine the local environments to an atomic level, was performed.” to “To reveal the presence of any ternary or amorphous compound other than 3D CsPbBr₃ and 0D Cs₄PbBr₆ that can be corroborated with the PXRD results, magic-angle-spinning (MAS) solid-state nuclear magnetic resonance (ssNMR), a sensitive characterization tool used to determine the local environments to an atomic level, was performed.” on page 11, line 205 in the revised manuscript.
3. We changed the following original sentence “Importantly, the presence of 2D CsPb₂Br₅, which was not confirmed by PXRD, was detected in the 20-min sample with its δ_{iso} at 235.7 ppm (see the semi-transparent purple strip in the inset of Fig. 3b).” to “Moreover, the presence of 2D CsPb₂Br₅ was detected only in the 20-min sample (i.e., in the early stage of the synthesis) with its δ_{iso} at 235.7 ppm (see the semi-transparent purple strip in the inset of Fig. 3b) and vanishes afterwards.” on page 12, line 218 in the revised manuscript.

References

- [R1] Saidaminov, M. I. et al. Pure Cs₄PbBr₆: Highly luminescent zero-dimensional perovskite solids. *ACS Energy Lett.* **1**, 840–845 (2016).
- [R2] De Bastiani, M. et al. Inside perovskites: Quantum luminescence from bulk Cs₄PbBr₆ single crystals. *Chem. Mater.* **29**, 7108–7113 (2017).
- [R3] Chen, Y.-M. et al. Cs₄PbBr₆/CsPbBr₃ perovskite composites with near-unity luminescence quantum yield: Large-scale synthesis, luminescence and formation mechanism, and white light-emitting diode application. *ACS Appl. Mater. Interfaces* **10**, 15905–15912 (2018).
- [R4] de Mello, J. C., Wittmann, H. F. & Friend, R. H. An improved experimental determination of external photoluminescence quantum efficiency. *Adv. Mater.* **9**, 230–232 (1997).
- [R5] Zhang, Z. et al. Growth, characterization and optoelectronic applications of pure-phase large-area CsPb₂Br₅ flake single crystals. *J. Mater. Chem. C* **6**, 446–451 (2018).

In addition, we would like to make following two remarks:

[Remark 1] – Modification on the paper title

We would like to change the title of our manuscript from “Designing highly luminescent Cs–Pb–Br composite perovskite through tracking the time evolution of mechanochemical synthesis” to “Mechanochemistry-driven engineering of 0D/3D heterostructure for designing highly luminescent Cs–

Pb–Br perovskites". We believe the latter conveys more concise and impactful impression considering the broad audience of *Nature Communications*, as well as better reflecting the main achievement made in this study.

[Remark 2] – Minor correction

In the inset of Fig. 3d, the Pb–Br configuration of PbBr_2 is referred to as "Edge & Corner-sharing heptahedra". Here, the word "heptahedra" should be changed to "**polyhedra**". On the same note, we changed the word "heptahedra" to "**polyhedra**" on page 14, line 257 in the revised manuscript. This modification does not alter any scientific claims in the manuscript.

In summary, we did our best to answer the reviewer's comments, and revised our manuscript appropriately. With these, we hope our manuscript can now be accepted for the publication in *Nature Communications*. Thank you.

=====
Responses to the Reviewer #2's comments
=====

[Overall evaluation]

I enjoyed reading your manuscript on the mechano-chemical synthesis of Cs–Pb–Br materials - in particular your detailed analysis of the evolution of materials formation. Indeed I find your work novel and refreshing as it provides some additional insight into the origin of green luminescence observed from what I agree are probably CsPbBr₃ inclusions in a higher bandgap matrix. Your correlative XRD and PL results show that the evolution of PL and XRD signatures indeed correlate and that there are also effects of changes in the PLQY to consider showing that PL is not the best quantitative probe of the amount of CsPbBr₃ formed. The manuscript is very well written and can in principle be published as-is. I would like the authors to consider the following suggestions for minor amendments or things to consider.

[Response]

We appreciate the reviewer's valuable time and efforts for evaluating our manuscript. In the following, we summarized the point-by-point responses and changes we have made on the basis of the reviewer's suggestions. Please note that we added additional remarks on page 11 of this response letter file suggesting a modification on the manuscript title and an additional minor correction that we found during this round of revision.

[Comment 1]

1. Consider to also add the PL and absorption spectra of samples before milling in Figure 2. This should of course be a fairly boring flat line but would confirm, that before milling no emissive species is present.

[Response]

We appreciate the reviewer's suggestion, the main concern of which is to confirm the absence of any emissive species in the 0-min sample (i.e., before any milling). As expected by the reviewer, no PL emission is observed from the 0-min sample (shown as flat baseline, Fig. R2a below). Although a small absorption onset of ~532 nm was measured from the absorption spectrum (Fig. R2b), the corresponding PLQE value of the 0-min sample is less than 0.1 % (nominal value calculated as 0.07 %), thus confirming the absence of any significant emission source in this sample (see Fig. R2c). For the sake of clarity, however, we decided to add these base-line spectra for the 0-min sample in the Supplementary Information instead of overcrowding the already densely packed main Figs. 2c and 2d.

Figure R2. **a**, PL spectra of 0-min and 5-min sample with an excitation wavelength of 365 nm. **b**, Absorption spectra of 0-min and 5-min sample. **c**, PL spectra of the 0-min sample for PLQE measurement with an excitation wavelength of 365 nm.

In response to the reviewer's comment, we added **Supplementary Fig. 3** along with the corresponding explanation in the revised Supplementary Information and added a sentence **“We note that sample before milling showed a near zero PLQE value of 0.07 %, indicating that no emissive properties were present beforehand (see Supplementary Fig. 3).”** on page 8, line 149 in the revised manuscript.

[Comment 2]

2. As noted by M Green and co-workers in a very comprehensive publication in 2015 (J. Phys. Chem. Lett. 2015, 6, 4774–4785), in materials with modest exciton binding energies, the excitonic absorption at the absorption onset obscures the actual bandgap. The bandgap cannot be directly determined from Tauc plots and authors should maybe consider to just refer to this as the absorption onset.

[Response]

Thank you for indicating this point. As the reviewer kindly explained, we agree that the bandgap of a material should indeed be carefully determined especially with materials of nontrivial exciton binding energies. We would like to use “optical bandgap” to refer to the reviewer's suggestion of “absorption onset”, for the sake of clarity. We note that what we are determining from the Tauc plot (Tauc model or Cody model [R6]) is the optical bandgap which is the value after excluding the exciton binding energy from the “bandgap” as defined by the fundamental transport gap [R7].

In response to the reviewer's comment, we revised our manuscript as following

1. We used the **“optical bandgap”** term when referring to the bandgap acquired through Tauc plot on page 9, line 157 and page 9, line 159 in the revised manuscript.
2. We clarified the caption of Fig. 2e **“e, Calculated optical bandgaps as a function of milling time obtained from the Tauc plot in Supplementary Fig. 5.”** from the original manuscript to **“e, Calculated optical bandgaps as a function of milling time. Values of optical bandgaps correspond to the absorption onsets of the Tauc plots⁶⁰ as shown in Supplementary Fig. 8.”** on page 38, line 619 in the revised manuscript.
3. We added the following paper as **Ref. 60** in the revised manuscript.

60. Green, M. A., Jiang, Y., Soufiani, A. M. & Ho-Baillie, A. Optical properties of photovoltaic organic–inorganic lead halide perovskites. *J. Phys. Chem. Lett.* **6, 4774–4785 (2015).**

References

- [R6] Raciti, R., Bahariqushchi, R., Summonte, C., Aydinli, A., Terrasi, A. & Mirabella, S. Optical bandgap of semiconductor nanostructures: Methods for experimental data analysis. *J. Appl. Phys.* **121**, 234304 (2017).
- [R7] Bredas, J.-L. Mind the gap! *Mater. Horiz.* **1**, 17–19 (2014).

In addition, we would like to make following two remarks:

[Remark 1] – Modification on the paper title

We would like to ask the reviewers' opinion about changing the title of our manuscript from “Designing highly luminescent Cs–Pb–Br composite perovskite through tracking the time evolution of mechanochemical synthesis” to “**Mechanochemistry-driven engineering of 0D/3D heterostructure for designing highly luminescent Cs–Pb–Br perovskites**” during this round of revision. We believe the latter conveys more concise and impactful impression considering the broad audience of *Nature Communications*, as well as better reflecting the key achievement made in the present study.

[Remark 2] – Minor correction

In the inset of Fig. 3d, the Pb–Br configuration of PbBr_2 is referred to as “Edge & Corner-sharing heptahedra”. Here, the word “heptahedra” should be changed to “**polyhedra**”. On the same note, we changed the word “heptahedra” to “**polyhedra**” on page 14, line 257 in the revised manuscript. We believe that this modification is not significant enough to alter any scientific claims in our previous submission.

In summary, we did our best to answer the reviewer's comments, and revised our manuscript appropriately. With these, we hope our manuscript can now be approved for the publication in *Nature Communications*. Thank you.

< End of the response letter >

REVIEWERS' COMMENTS

Reviewer #1 (Remarks to the Author):

The authors have addressed my comments very well and the manuscript has been improved. In my opinion this work can be accepted now.

Reviewer #2 (Remarks to the Author):

Dear Authors,

I am content with your responses and edits to your manuscript.

=====
Responses to the Reviewer #1's comments
=====

[Overall evaluation]

The authors have addressed my comments very well and the manuscript has been improved. In my opinion this work can be accepted now.

[Response]

We appreciate the reviewer's valuable time and efforts for evaluating our revised manuscript.

=====
Responses to the Reviewer #2's comments
=====

[Overall evaluation]

I am content with your responses and edits to your manuscript.

[Response]

We appreciate the reviewer's valuable time and efforts for evaluating our revised manuscript.

< End of the response letter >